*Perspective*

EMBO
Molecular Medicine

# Pertussis before, during and after Covid-19

Camille Locht ✉

## Abstract

After the Covid-19 pandemic, pertussis has made a spectacular comeback in Europe and many other parts of the world, while during the pandemic it had essentially disappeared because of the social distancing requirements. However, even before the Covid-19 pandemic, the disease was on the rise in many countries, especially those that have replaced whole-cell pertussis vaccines by acellular pertussis vaccines. Several reasons may account for this upsurge, including strain adaptation to escape vaccine-induced immunity, rapid waning of immunity after vaccination and the failure of current vaccines to prevent infection by and transmission of the causative agent *Bordetella pertussis*. Various strategies have been deployed to control the disease, the most effective of which is maternal vaccination during pregnancy to protect the newborn against the most severe and deadly forms of the disease. However, ultimate control of pertussis likely requires novel vaccines, which prevent infection and transmission, not only disease. One of them is the live attenuated BPZE1 vaccine, which has shown promise in pre-clinical and clinical studies and may therefore perhaps become a gamechanger.

**Keywords** Whooping Cough; Whole-cell Vaccines; Acellular Vaccines; Mucosal Immunity

Pertussis or whooping cough is a vaccine-preventable, highly contagious respiratory disease, mainly caused by the Gram-negative bacterium *Bordetella pertussis*. It affects all age groups, but is particularly severe and life-threatening in young infants. Since the introduction of large-scale vaccination programs, starting in the late 1950s, initially with first-generation, whole-cell pertussis vaccines (wPV), and since the early 2000s with second-generation, acellular pertussis vaccines (aPV), pertussis incidence has plummeted. Pertussis vaccines are generally used as combined vaccines, including diphtheria and tetanus valencies, as well as sometimes additional valencies. Most pertussis vaccine regimens include 3 primary doses given in early childhood, usually starting at 6 weeks or 2 months of age and being completed at 6 months of age. A fourth dose is sometimes included later in life.

## Pertussis during and after the Covid-19 pandemic

During the Covid-19 pandemic, pertussis has almost completely disappeared in many countries (European Center for Disease Prevention and Control, 2024), most likely due to the non-pharmaceutical anti-Covid-19 interventions, such as wearing face masks, social distancing and confinement, deployed during that time. In Europe no infant death due to pertussis was reported during the Covid-19 pandemic. However, several pertussis-linked deaths have occurred in the elderly during that time, most of which were associated with underlying conditions, such as asthma, chronic obstructive pulmonary disease and immunosuppression. It is not known whether SARS-CoV2 infection itself may have exacerbated pertussis during co-infection, especially in the elderly.

Curiously, when counter-epidemic measures were relaxed, pertussis incidences remained low during the initial months, while other respiratory infections, especially the Respiratory Syncytial Virus infections, skyrocketed, causing a first wave of post-Covid-19 hospital admissions. However, the first half of 2024 has witnessed a spectacular rebound in pertussis incidence in Europe (European Center for Disease Prevention and Control, 2024), with peak incidence rates in Denmark of more than 330/100,000, not seen since over 20 years (Nordholm et al, 2024), and in the Czech Republic the largest outbreak in more than 60 years (Holt, 2024). Other parts of the world, such as China (Hu et al, 2024), also experienced massive pertussis outbreaks, with numbers of cases often largely exceeding those in the pre-Covid-19 era.

The reasons for this resurgence are not yet completely elucidated. The Covid-19 pandemic has not led to a significant drop in vaccine coverage, at least not in Europe, where vaccination coverage for the three primary vaccine doses consistently ranged between 94 and 96% since 2012 (European Center for Disease Prevention and Control, 2024). It is thus unlikely that the recent resurgence may be due to insufficient vaccination coverage. Interestingly, however, the age distribution of pertussis cases has shifted from the less than 1-year-old infants before the Covid-19 pandemic to the 10–19 years-old children as the major age category of pertussis cases in 2024.

Although in the post-Covid-19 era, pertussis incidences soared spectacularly, pertussis cases were already on the rise before the Covid-19 pandemic. This was especially the case in countries that have replaced the wPV by aPV (Esposito et al, 2019).

## Reasons for the upsurge of pertussis in countries using aPV

Besides improved technologies to detect pertussis cases, such as introduction of multiplex polymerase-chain-reaction panels for bacterial respiratory infection diagnosing, and enhanced awareness among physicians and caretakers, which might have contributed to increased pertussis reporting, three major shortcomings of current vaccines are likely to be the major drivers of the pertussis upsurge (Esposito et al, 2019). Current aPV are composed of 2–5 purified

Univ. Lille, CNRS, Inserm, CHU Lille, Institut Pasteur de Lille, U1019-UMR9017–CIIL—Centre for Infection and Immunity of Lille, Lille, France.
✉E-mail: Camille.locht@pasteur-lille.fr
https://doi.org/10.1038/s44321-025-00199-2 | Published online: 24 February 2025

*B. pertussis* antigens. All vaccines contain detoxified pertussis toxin (PT) and filamentous haemagglutinin (FHA). Most contain pertactin in addition, and some further contain serotype 2 and serotype 3 fimbriae. Among those antigens only pertactin induces bactericidal antibodies (Lesne et al, 2020), and since the introduction of pertactin-containing aPV, many breakthrough infections were due to pertactindeficient strains. In fact, there is a strong link between pertactin deficiency and the time of aPV introduction (Barkoff et al, 2019). The molecular mechanisms that led to pertactin deficiencies in these strains are very diverse and include gene deletions, transposon insertions, frameshift mutations and promoter inversions. These observations strongly suggest strain adaptation of *B. pertussis* to escape vaccine pressure as one of the reasons for pertussis upsurge.

A second reason is the fast waning of aPV-induced immunity. This was initially noticed in Australia, where the transition from wPV to aPV started in 1999. Therefore, children born in 1998 had received for the three-dose primary vaccination either exclusively wPV, aPV or a mix of the two. During the 2009–2011 pertussis outbreak, substantially more pertussis cases were reported among children who had received the aPV than among the wPV recipients (Sheridan et al, 2012). Studies in other countries, such as Canada, found that aPV effectiveness dropped from approximately 80% during the first year after vaccination to 60% 3 years and less than 40% five years after vaccination (Schwartz et al, 2016).

The third shortcoming of current aPV is their failure to prevent infection by and transmission of *B. pertussis*. Pertussis is, together with measles, among the two most contagious diseases (Anderson and May, 1982), and whereas measles is well controlled by vaccination, provided the coverage is high enough, this is not the case for pertussis. In the first half of the 20th century, before vaccination was implemented, both measles and pertussis occurred in cycles of 3–5 years. After the successful introduction of vaccines against these diseases, both measles and pertussis incidences declined markedly. However, whereas for measles the intervals between epidemic peaks increased substantially, the interpeak intervals did not appreciably increase for pertussis. These epidemiological features reflect the fact that measles vaccines prevent both disease from and transmission of the measles virus and induce herd immunity, which is not the case for pertussis vaccines. Studies in non-human primates have confirmed that current pertussis vaccines, while highly protective against pertussis disease, do not prevent infection by *B. pertussis* (Warfel et al, 2014). In addition, these studies have shown that aPV-treated animals are able to transmit *B. pertussis* as effectively as non-vaccinated animals. Furthermore, studies in murine models have shown that immunization with aPV prolongs nasal carriage by inhibiting the induction of naturally induced IL-17-producing resident memory T ($T_{RM}$) cells (Dubois et al, 2021), which are essential for the recruitment of neutrophils able to kill the bacteria in the nasal cavity (Allen et al, 2018; Borkner et al, 2021). If these nonclinical studies can be extrapolated to humans, the introduction of aPV may have considerably increased the *B. pertussis* reservoir, which may have contributed to the resurgence of pertussis. A recent study has demonstrated that human adults primed in their childhood with aPV had significantly less *B. pertussis*-specific IL-17-producing $T_{RM}$ cells in their nasal tissues than those that had been primed with wPV (McCarthy et al, 2024).

All three limitations of current aPV likely play a role in the present-day epidemiological pertussis situation. However, using wavelet analyses of pertussis incidences in the United States and the United Kingdom, together with phylogenetic analyses of clinical *B. pertussis* isolates combined with mathematical transmission models Althouse and Scarpino (2015) concluded that asymptomatic transmission was in fact the main driver of the increased pertussis incidence, at least in these two countries.

## Proposed solutions

Although pertussis affects all age groups, the most vulnerable ones are young infants below 6 months of age, especially those that have not completed the primary vaccination course. This is also the age group with the highest mortality rates. Therefore, all prevention strategies primarily target the newborns.

With the assumption that the source of transmission in cases of infant pertussis is for more than 70% a household contact, mostly a parent or a sibling, but also including grandparents, uncles and aunts, it was hypothesized that vaccination of close contacts with a newborn would substantially reduce the numbers of infant pertussis cases. Consequently, maternal, family and household vaccination, also referred to as "cocooning", has been recommended in several countries since the early 2000s, and is today still recommended in some countries, including France. However, besides the difficulties in implementing cocooning strategies to achieve sufficient coverage, studies conducted in settings where cocooning was successfully implemented have shown that it did not reduce pertussis illness in infants less than 6 months of age (Healy et al, 2015). Today we know that the reason for the ineffectiveness of cocooning newborns is the inability of aPV to prevent infection and transmission.

Since the early 2010s, pertussis immunization during pregnancy is being recommended in several high-income countries, with the premise that antibodies generated by vaccination during pregnancy would be actively transferred via the placenta to the fetus, so that the newborn may be protected against disease from birth on. The effectiveness of this approach has been clearly demonstrated during the 2012 pertussis outbreak in the United Kingdom when the UK Department of Health recommended maternal pertussis vaccination during pregnancy. Based on a coverage rate of 60%, vaccine effectiveness could be estimated at 91%, provided that the mothers were vaccinated at least 7 days before birth. Vaccination closer to delivery or shortly after delivery resulted in a substantial dop in effectiveness (Amirthalingam et al, 2014). Numerous other studies have since then confirmed high effectiveness of maternal vaccination during pregnancy against early childhood pertussis, and this strategy is today regarded as the most efficient approach to protect neonates against severe and life-threatening pertussis (Sadarangani et al, 2021). Maternal vaccination is therefore now recommended at each pregnancy in several countries.

However, many studies have also shown that pertussis vaccination during pregnancy results in blunting of the immune responses to primary vaccination, especially when wPV are used for the primary course (Abu-Raya et al, 2021). Anti-PT and anti-FHA serum antibody levels upon wPV immunization were significantly lower in children born to vaccinated mothers than in children born to unvaccinated mothers, and

these differences were maintained after booster doses at 18 months (Wanlapakorn et al, 2020). Similar, albeit less pronounced blunting effects have also consistently been seen when aPV were used for the primary vaccination course. It is not known whether this blunting effect has any clinical consequences, and as of today, there are no signs of reduced effectiveness of primary vaccination due to maternal immunization. On the other hand, murine studies have shown that maternal aPV immunization results in prolonged nasal carriage of *B. pertussis* in the offspring (Dubois et al, 2023). Therefore, it cannot be ruled out that maternal vaccination programs will ultimately result in an increase of the *B. pertussis* reservoir, which, considering the high infectivity of *B. pertussis*, may in the future potentially further complicate control strategies for pertussis.

## New pertussis vaccines

In recent years it has become increasingly apparent that in order to ultimately control pertussis at a population level, vaccines need to not only prevent disease but must also prevent infection and transmission. As this may be very difficult, if not impossible to achieve with the current vaccines, new vaccines need to be developed. Studies on the mechanism of anti-pertussis immunity have highlighted the importance of mucosal immune mechanisms, both at the humoral and cellular level, for protection against *B. pertussis* infection (reviewed in Dubois and Locht, 2021). Especially cell-mediated immunity plays an important role in vaccine effectiveness and duration of immunity (Redhead et al, 1993). In contrast to natural infection, aPV induces a predominantly T helper 2 (Th2) type immunity (Mascart et al, 2003) at the expense of Th1 and Th17 responses, which are associated with improved and prolonged protection (Allen et al, 2018). In mice aPV was shown to suppress naturally induced local Th17 responses in lungs and noses via a mechanism involving IL-4 signaling (Dubois et al, 2023).

Consequently, much effort has been deployed to change the aPV-induced cytokine profile by formulating the vaccines with a variety of different adjuvants. Murine studies have shown that orientation of cytokine responses to aPV into the desired direction via the use of various adjuvants is possible (Misiak et al, 2017). However, even with these formulations the impact of the vaccines on nasal colonization by *B. pertussis* was

generally minimal when the vaccines were injected systemically. It is commonly accepted that the optimal route for the induction of mucosal immunity in the respiratory tract is the intranasal route. However, in order to achieve potent mucosal immunity to nasally administered aPV, mucosal adjuvants are needed, yet no safe mucosal adjuvant for nasal administration in humans is presently available. Nevertheless, various adjuvant formulations, sometimes in combination with lipid nanoparticles, given intranasally have shown promise in mice for the induction of mucosal IgA responses and IL-17-producing $T_{RM}$ cells, as well as decreased nasal colonization by *B. pertussis* upon challenge (reviewed in Dubois and Locht, 2021).

An alternative approach intensively pursued is the use of outer membrane vesicles (OMVs). This approach has the advantage of broadening the antigen repertoire compared to aPV, which would be expected to minimize the risk of generating vaccine escape *B. pertussis* strains, as has been documented after the use of aPV by the emergence of pertactin-deficient strains. Nasal or pulmonary administration of *B. pertussis* OMVs indeed generated mucosal antibody responses with broad specificity, local Th1/Th17 memory T cell responses, as well as significant protection against lung and nasal colonization by *B. pertussis* in mice (Raeven et al, 2020).

## The live attenuated BPZE1 vaccine candidate

Although many novel vaccine candidates have been assessed in mouse models, only the live attenuated vaccine candidate BPZE1 has also been evaluated in non-human primates and in humans. This vaccine candidate was designed to mimic natural infection without causing disease, since natural infection has been well-established to induce strong protection against reinfection and longer protection against disease than vaccination (Wendelboe et al, 2005).

BPZE1 was constructed based on the existing knowledge on molecular mechanisms of pertussis pathogenesis and *B. pertussis* virulence. Adhesins, such as FHA, fimbriae and others were maintained, while several toxins were genetically removed or inactivated (Mielcarek et al, 2006). The strain lacks the dermonecrotic toxin gene, coding for a highly toxic protein, has undetectable levels of tracheal cytotoxin, a peptidoglycan breakdown product that can destroy ciliated epithelial cells,

and has a genetically modified analog of PT. The latter toxin expresses enzymatic ADP-ribosyltransferase activity, which is the molecular basis of PT intoxication. Pertussis toxoid is also an essential component of all pertussis vaccines, because antibodies that neutralize PT have been shown to protect against severe disease. It is therefore important that pertussis vaccination induces PT-neutralizing antibodies. In BPZE1 PT is genetically inactivated by the replacement of two amino acids responsible for two independent steps in catalysis, which makes reversion virtually impossible.

Pre-clinical studies, initially in mice, have shown that BPZE1 is safe, even at very high doses, including in severely immunocompromised and neonatal mice, and that it is genetically stable in vitro and in vivo after several passages. A single nasal administration of BPZE1 induced a strong serum and mucosal antibody response, including secretory IgA (sIgA), as well as a Th1-type cytokine profile. Importantly, it also induced $T_{RM}$ cells that produce IL-17, IFN-γ or both in the nasal tissues (Solans et al, 2018). Especially, the nasal IL-17-producing $T_{RM}$ cells were shown to be an important contributor to long-lasting immunity in the upper respiratory tract (Allen et al, 2018).

In juvenile baboons, a single administration of BPZE1 was also found to be safe and to prevent *B. pertussis*-induced leukocytosis, a hallmark of severe pertussis disease. BPZE1 vaccination also substantially reduced the bacterial burden in the nasal cavity of the baboons challenged with a very high dose of a highly pathogenic *B. pertussis* clinical isolate (Locht et al, 2017). High levels of serum antibodies, both *B. pertussis*-specific IgG and IgA, were also induced by a single BPZE1 administration, which were further increased after virulent challenge.

BPZE1 has entered clinical development in the early 2010s, first as a liquid formulation stored at −80 °C, and more recently as a more stable, lyophilized formulation (Thalen et al, 2020). A first-in-human, randomized, placebo-controlled, double-blind clinical trial showed that the vaccine was able to transiently colonize the human nasopharynx and was safe up to a dose of at least $10^7$ colony-forming units (CFU) in healthy human male adults (Thorstensson et al, 2014). This dose-escalation study, with doses ranging from $10^3$, over $10^5$ to $10^7$ CFU, also showed that the colonization frequency depended on the dose, reaching about 40% of the study subjects being colonized at the

highest dose tested in this study. All subjects colonized with the highest dose also mounted serum antibody responses to PT, FHA, fimbriae, and pertactin, which remained stable for at least up to 6 months after vaccination. However, not all participants were colonized, and those that were not colonized had significantly higher baseline serum antibody levels compared with those that were colonized, suggestive of prior infection by *B. pertussis*, which may have protected against attenuated challenge infection. In a second phase I trial, volunteers with high pre-existing antibody levels to *B. pertussis* antigens were excluded, and the doses were increased from $10^7$, over $10^8$ to $10^9$ CFU BPZE1 (Jahnmatz et al, 2020). In this study colonization reached ~80% in each of the three groups, and in the highest-dose group, all subjects seroconverted.

In a subsequent phase II trial with the $10^9$ CFU dose, BPZE1 was compared to aPV (Keech et al, 2023). As expected, aPV induced a strong serum antibody response to *B. pertussis* antigens, but did not induce a significant mucosal sIgA response and did not prevent infection by a second BPZE1 dose, used as an attenuated challenge. In contrast, intranasal immunization with BPZE1 resulted in a strong *B. pertussis*-specific nasal sIgA response, in addition to serum antibody responses, and resulted in a substantial reduction in bacterial load after a second BPZE1 administration. These observations thus provided proof of concept that nasal vaccination with BPZE1 may prevent infection by *B. pertussis* in humans.

A follow-up study, using a controlled human infection model with virulent *B. pertussis* [NCT05461131], confirmed these conclusions and showed that BPZE1 can also protect against infection by virulent *B. pertussis* in humans. Currently, BPZE1 is also being evaluated for safety and immunogenicity in school-age children between 6 and 17 years of age [NCT05116241]. They had all been primed in their early childhood with aPV, in contrast to the adults included in the previous studies, most of whom had been primed with wPV. Studies on aPV-primed individuals are particularly important, as animal studies have revealed that aPV priming may blunt IL-17 production by TRM cells (Dubois et al, 2021; McCarthy et al, 2024), and work with non-human primates has shown that multiple *B. pertussis* infections may be required to overcome this blunting of aPV priming (Kapil et al, 2024). A two-dose BPZE1 regimen may therefore perhaps be necessary to fully override IL-17 blunting in aPV-primed subjects. The vaccine has now been administered to more than 600 human subjects, children, adolescents and adults, without any safety signal. If a regulatory path can be found to bring it to the market, this vaccine holds thus promise as a tool for effective control of pertussis.

## Peer review information

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

## Author contributions

**Camille Locht**: Conceptualization; Writing—original draft; Writing—review and editing.

## Disclosure and competing interests statement

Camille Locht is the co-inventor of BPZE1, and its employer holds a patent folio on the technology, currently licensed to ILiAD Biotechnologies. He also reports consulting fees from, and holds equity in ILiAD Biotechnologies.

