## [Peer Review File · EMBO Molecular Medicine]

Pertussis before, during and after Covid-19

Camille Locht

Corresponding author: Camille Locht (camille.locht@pasteur-lille.fr)

Review Timeline:

Submission Date:	4th Dec 24
Editorial Decision:	9th Jan 25
Revision Received:	15th Jan 25
Editorial Decision:	21st Jan 25
Revision Received:	22nd Jan 25
Accepted:	23rd Jan 25

Editor: Lise Roth

Transaction Report:

9th Jan 2025

Dear Dr. Locht,

Thank you for submitting your commentary to EMBO Molecular Medicine, and please accept my apologies for the delay in getting back to you in this busy time of the year. I have now received the feedback from the reviewer who evaluated your manuscript. As you will see below, this expert is overall positive, but nevertheless has minor suggestions.

We will thus welcome the submission of a revised version of your article that would satisfactorily address the referee's concerns. In particular, he/she suggests balancing the references to cite more work from other experts in the field.

Based on this recommendation and on the current format of your piece, we would suggest changing the type of article to "Perspective", recently introduced in EMM:

"A Perspective article should 'set the scene' based on recent developments with an emphasis on future directions of the field of study. Perspectives can take a personal point of view, but should emphasize reported facts and testable hypotheses over speculation and opinion. The format is flexible, but we aim for succinct and focussed discussions to engage a broad readership."

No particular changes are required, but it would give you more flexibility regarding text length and number of references.

When submitting your revised manuscript, please carefully review the instructions that follow below. We require:

- 1) A .docx formatted version of the manuscript text. Please make sure that the changes are highlighted to be clearly visible.
- 2) If applicable: Individual production quality figure files as .eps, .tif, .jpg (one file per figure).
- 3) An ORCID identifier is required.
- 4) The format of the references should be changed from numerical to alphabetical.
- 5) As part of the EMBO Publications transparent editorial process initiative (see our Editorial at <http://embomolmed.embopress.org/content/2/9/329>), EMBO Molecular Medicine will publish online a Review Process File (RPF) to accompany accepted manuscripts.

This file will be published in conjunction with your paper and will include the anonymous referee reports, your point-by-point response and all pertinent correspondence relating to the manuscript. Let us know whether you agree with the publication of the RPF.

I look forward to receiving your revised manuscript.

With kind regards,

Lise Roth

***** Reviewer's comments *****

Referee #1 (Remarks for Author):

The succinct minireview by Dr. Locht is rather straightforward and deals with an important topic and timely topic of pertussis resurgence. However, compared to previous texts of the author, it brings a limited amount of novel insight. Nobody questions the contribution of Dr. Locht to the field of pertussis research and vaccines, but the extensive self-citation should be balanced by addition of more references to the pioneering work of others, as well.

Some language improvements are necessary and a critical issue of overcoming of blunting of aPV-elicited block of IFN γ /IL-17A-secreting T cells by BPZE1 infection needs to be discussed in the light of recent results of Kapil et al. 2023 doi:

10.1093/infdis/jiad332 (see below).

Here are my minor suggestions:

- I. 30 - ...which prevent infection and transmission NOT ONLY disease.
- I. 40 - ...acellular PERTUSSIS vaccines (aPV), pertussis incidence HAS plummeted.
- I. 44 - A FOURTH dose...
- I. 47 - ...pertussis HAS...disappeared...
- I. 49 - ...interventions, such as wearing of face masks, social distancing and confinement, deployed...
- I. 55 - ...when counter-epidemic measures...
- I. 56 - ...especially THE Respiratory...
- I. 70 - in the epidemics in Denmark, Czech Republic and some other countries it was rather the 15 - 19 years old adolescent group that was the major driver of the outbreak
- I. 77 - ...improved diagnostic technologies to detect pertussis cases, such as introduction of multiplex PCR panels for bacterial respiratory infection diagnosing, and enhanced awareness...
- I. 91 - sentence needs improvement: This was initially noticed in Australia,...
- I. 94 - The Australian outbreak started already in 2009: ... During the 2009-2011 pertussis outbreak...
- I. 108 - ...prevent both disease FROM and TRANSMISSION OF the measles virus... immunity, this is...
- I. 115 - the self-citation here is exaggerated: insert references to at least some papers of the Mills group that discovered the role of Trm cells producing IL-17A and attracting Siglec F+ neutrophils to clear the infection (Misiak 2017, Wilk 2017, Allen 2018, Wilk 2019, Borkner 2021) in protection of mucosa from infection and colonization by B. pertussis.

- I. 118 - add discussion and reference of the data from humans by McCarthy et al. J Infect Dis. 2024 Sep 23;230(3):e518-e523. doi: 10.1093/infdis/jiae034, who shows that aPV-primed and boosted adult humans have importantly less resident memory T cells reacting to B. pertussis antigens by IL-17-secretion than do wPV-primed adults have = this is highly relevant here and validates the animal models data.

- I. 132 - ...parent, OR SIBLING, but also grandparents, uncles...
- I. 179-184 - the work of Dubois et al. 2023 is certainly of importance, but the role of cell mediated immunity in vaccine protection was revealed already in 1993 by the works of Mills and Readhead (2 papers in Infect. Immun.) and also the above mentioned seminal papers of Mills group need to be somehow cited here
- I. 204 - reference missing - add
- I. 219 - PT ANTIGEN (or toxoid) is also ...
- I. 235 - reference to the baboon study missing - add
- I. 241 - NASOPHARYNX
- I. 265 and 267 - sIgA (not S-IgA)

- I. 279 - 287 - The BPZE1 concept is surely very promising but the "efficacy" of the BPZE1 vaccine was only shown so far in wPV-primed adults that were protected against subsequent virulent B. pertussis challenge. A major bottleneck in proving efficacy in aPV-primed individuals, which are the source of the pertussis problem to be tackled, is the observed blunting of IFN γ and IL-17 responses of T cells of aPV-primed and boosted baboons, as reported by Kapil et al. 2023, Repeated Bordetella pertussis Infections Are Required to Reprogram Acellular Pertussis Vaccine-Primed Host Responses in the Baboon Model. J Infect Dis. 2024 Feb 14;229(2):376-383. doi: 10.1093/infdis/jiad332.
Given that aPV-primed humans have significantly less resident memory T cells and repeated high dose infection of baboon did not fully override the aPV-induce blunting of IL-17/IFN γ -polarized T cell responses, it needs to be shown that BPZE1 infection will trigger protective resident memory T cells producing IL-17 to confer longer-lasting protection, aside of sIgA induction. This issue should not be neglected and needs to be discussed at the end of the review, in the outlook paragraph, as it may represent a major obstacle of clinical efficacy of the BPZE1 vaccine in the targeted aPV-primed and boosted humans.

I have addressed the various comments as follows:

The succinct minireview by Dr. Locht is rather straightforward and deals with an important topic and timely topic of pertussis resurgence. However, compared to previous texts of the author, it brings a limited amount of novel insight. Nobody questions the contribution of Dr. Locht to the field of pertussis research and vaccines, but the extensive self-citation should be balanced by addition of more references to the pioneering work of others, as well.

Reply: In the revised version I have added more references to the work others. This is now possible due to the change of article type to “Perspective”.

Some language improvements are necessary and a critical issue of overcoming of blunting of aPV-elicited block of IFN γ /IL-17A-secreting T cells by BPZE1 infection needs to be discussed in the light of recent results of Kapil et al. 2023 doi: 10.1093/infdis/jiad332 (see below).

Reply: In the revised version, I now discussed the very relevant work of Kapil et al. and of course cited this reference (see the reply to the last comment below).

Here are my minor suggestions:

l. 30 - ...which prevent infection and transmission NOT ONLY disease.

Reply: This change was made in the revised version

l. 40 - ...acellular PERTUSSIS vaccines (aPV), pertussis incidence HAS plummeted.

Reply: This change was made in the revised version

l. 44 - A FOURTH dose...

Reply: This change was made in the revised version

l. 47 - ...pertussis HAS...disappeared...

Reply: This change was made in the revised version

l. 49 - ...interventions, such as wearing of face masks, social distancing and confinement, deployed...

Reply: This has been added in the revised version

l. 55 - ...when counter-epidemic measures...

Reply: This change was made in the revised version

l. 56 - ...especially THE Respiratory...

Reply: This change was made in the revised version

l. 70 - in the epidemics in Denmark, Czech Republic and some other countries it was rather the 15 - 19 years old adolescent group that was the major driver of the outbreak

Reply: This may be true for some countries, like Denmark and the Czech Republic. However, according to the eCDC (European Center for Disease Prevention and Control. 2024. Increase of pertussis cases in the EU/EEA, 8 May 2024. Stockholm change was made in the revised version) the major increase in Europe overall was seen in the 10-14 years old. However, I agree that there was also a strong increase in the 15-19 years old adolescents. I therefore changed to 10-19.

l. 77 - ...improved diagnostic technologies to detect pertussis cases, such as introduction of multiplex PCR panels for bacterial respiratory infection diagnosing, and enhanced awareness...

Reply: This has been added in the revised version

l. 91 - sentence needs improvement: This was initially noticed in Australia,...

Reply: This change was made in the revised version

l. 94 - The Australian outbreak started already in 2009: ... During the 2009-2011 pertussis outbreak...

Reply: This change was made in the revised version

l. 108 - ...prevent both disease FROM and TRANSMISSION OF the measles virus... immunity, this is...

Reply: This change was made in the revised version

l. 115 - the self-citation here is exaggerated: insert references to at least some papers of the Mills group that discovered the role of Trm cells producing IL-17A and attracting Siglec F+ neutrophils to clear the infection (Misiak 2017, Wilk 2017, Allen 2018, Wilk 2019, Borkner 2021) in protection of mucosa from infection and colonization by B. pertussis.

Reply: Although not all, the most important papers from the Mills group have been added in the revised version on several occasions throughout the article.

l. 118 - add discussion and reference of the data from humans by McCarthy et al. J Infect Dis. 2024 Sep 23;230(3):e518-e523. doi: 10.1093/infdis/jiae034, who shows that aPV-primed and boosted adult humans have importantly less resident memory T cells reacting to B. pertussis antigens by IL-17-secretion than do wPV-primed adults have = this is highly relevant here and validates the animal models data.

Reply: This discussion and the reference McCarthy et al have been added in the revised version.

l. 132 - ...parent, OR SIBLING, but also grandparents, uncles...

Reply: This has been added in the revised version

l. 179-184 - the work of Dubois et al. 2023 is certainly of importance, but the role of cell mediated immunity in vaccine protection was revealed already in 1993 by the works of Mills and Readhead (2 papers in Infect. Immun.) and also the above mentioned seminal papers of Mills group need to be somehow cited here

Reply: These references have been added in the revised version

l. 204 - reference missing - add

Reply: A reference has been added in the revised version

l. 219 - PT ANTIGEN (or toxoid) is also ...

Reply: I have been “toxoid” in the revised version

l. 235 - reference to the baboon study missing - add

Reply: A reference has been added in the revised version

l. 241 - NASOPHARYNX

Reply: This change was made in the revised version

l. 265 and 267 - sIgA (not S-IgA)

Reply: This change was made in the revised version

l. 279 - 287 - The BPZE1 concept is surely very promising but the "efficacy" of the BPZE1 vaccine was only shown so far in wPV-primed adults that were protected against subsequent virulent B. pertussis challenge. A major bottleneck in proving efficacy in aPV-primed individuals, which are the source of the pertussis problem to be tackled, is the observed blunting of IFN γ and IL-17 responses of T cells of aPV-primed and boosted baboons, as reported by Kapil et al. 2023, Repeated Bordetella pertussis Infections Are Required to Reprogram Acellular Pertussis Vaccine-Primed Host Responses in the Baboon Model. J Infect Dis. 2024 Feb 14;229(2):376-383. doi: 10.1093/infdis/jiad332.

Given that aPV-primed humans have significantly less resident memory T cells and repeated high dose infection of baboon did not fully override the aPV-induce blunting of IL-17/IFN γ -polarized T cell responses, it needs to be shown that BPZE1 infection will trigger protective resident memory T cells producing IL-17 to confer longer-lasting protection, aside of sIgA induction. This issue should not be neglected and needs to be discussed at the end of the review, in the outlook paragraph, as it may represent a major obstacle of clinical efficacy of the BPZE1 vaccine in the targeted aPV-primed and boosted humans.

Reply: I agree with the reviewer and have added a discussion on this issue at the end of the paper.

The corresponding changes in the revised version can be seen in track change mode.

21st Jan 2025

Dear Dr. Locht,

Thank you for submitting your revised Perspective to EMBO Molecular Medicine. I appreciate the revisions that have been done, and I further suggest some edits for the overall balance of the manuscript. Please let me know if you agree or amend as you see fit (document attached).

Once you have gone through the edits, please submit your final version for acceptance. Kindly let us know if you agree with the publication of the Review Process File (RPF). This file will be published in conjunction with your paper and will include the anonymous referee report, your point-by-point response and all pertinent correspondence relating to the manuscript.

Looking forward to receiving your revised manuscript.

With kind regards,

Lise Roth

The authors addressed the remaining formatting issues.

23rd Jan 2025

Dear Dr. Locht,

Thank you for submitting your revised file. I am pleased to inform you that your manuscript is now accepted for publication in EMBO Molecular Medicine!

Your manuscript will be processed for publication by EMBO Press. It will be copy edited and you will receive page proofs prior to publication. Please note that you will be contacted by Springer Nature Author Services to complete licensing information. We will soon send you a token that will allow you to be exempted from publication charges.

If you have any questions, please do not hesitate to contact the Editorial Office.

Thank you very much for your contribution to EMBO Molecular Medicine!

With kind regards,

Lise Roth
